# Extracting Work Optimally with Imprecise Measurements

**DOI:** 10.3390/e23010008

**Published:** 2020-12-23

**Authors:** Luis Dinis, Juan Manuel Rodríguez Parrondo

**Affiliations:** 1Grupo Interdisciplinar de Sistemas Complejos, Facultad de Ciencias Físicas, 28040 Madrid, Spain; parrondo@ucm.es; 2Departamento de Estructura de la Materia, Física Térmica y Electrónica, Universidad Complutense de Madrid, 28040 Madrid, Spain

**Keywords:** confinement, information theory, Brownian particle, stochastic thermodynamics

## Abstract

Measurement and feedback allows for an external agent to extract work from a system in contact with a single thermal bath. The maximum amount of work that can be extracted in a single measurement and the corresponding feedback loop is given by the information that is acquired via the measurement, a result that manifests the close relation between information theory and stochastic thermodynamics. In this paper, we show how to reversibly confine a Brownian particle in an optical tweezer potential and then extract the corresponding increase of the free energy as work. By repeatedly tracking the position of the particle and modifying the potential accordingly, we can extract work optimally, even with a high degree of inaccuracy in the measurements.

## 1. Introduction

Modern techniques have allowed for the manipulation of objects at the microscale. A paradigmatic example are colloidal particles trapped by optical tweezers. At this scale—the scale of Brownian motion—not only the motion of particles, but the energy fluxes, work, or heat, become stochastic. Nevertheless, the combination of manipulation and imaging or other detection techniques allow for some degree of control [1]. For instance, in driven systems, the external driving may be modified based on outcomes of measurements, as in feedback control, leading, for example, to (efficient) confinement in small region of space [2] or to the reduction of thermal fluctuations, i.e., cooling, a technique that is implemented in both classical or quantum systems [3,4]. Another application of feedback is an increase of the performance of certain motors operating at the microscale, such as Brownian ratchets or micro-motors [5,6,7,8,9].

Feedback exploits the information that is acquired through measurement as a thermodynamic resource. It is now known that the work needed to perform an isothermal feedback process, for a system in contact with an environment at constant temperature *T*, is bounded by the following extension of the second law of thermodynamics [9,10]:(1)W≥ΔF−kTI,
where ΔF is the free energy difference between the final and initial states of the process, *k* the Boltzmann’s constant, and *I* is the amount of information that is gained in the measurement, quantified by the mutual information from information theory. Information is always positive (or zero) and, thus, in a cycle (ΔF=0) it is possible to extract work (W<0) from a single thermal bath with measurement and feedback.

Equation (Equation 1) also shows that a given level of accuracy in the measurement, quantified by the mutual information, limits the amount of work that can be extracted in one feedback operation. Some especially tailored protocols saturate that bound (Equation 1) and they may be used to convert all of the information acquired into useful work. These are processes that are reversible under feedback [11,12,13]. In this article, we first review these protocols and show how to use their special properties in order to extract energy with the same efficiency and even power when operating with higher measurement errors. In order to fix ideas, we use a well known model that we proceed to describe in the following section.

## 2. Model Description and Cycle Operation

As our system, we consider an overdamped Brownian particle that is in contact with a thermal bath, which acts as its environment. The particle feels a harmonic potential. This is a well proven theoretical model for an experimental system that was formed by a colloidal particle in water at constant temperature and trapped by optical tweezers. The potential $V_{\kappa ,x_{0}}=\frac{1}{2}\kappa {\left(x-x_{0}\right)}^{2}$ has tunable parameters $x_{0}$, the position of the center of the trap, and κ, the stiffness. As the Brownian particle position fluctuates, the energy transfers and thermodynamic potentials become also fluctuating; in fact, they are stochastic variables. The framework to analyze energetics for these fluctuating systems in the mesoscale is stochastic thermodynamics [14,15,16]. We review the main concepts in the following. The Brownian particle may, due to a collision with the solvent, absorb some energy and climb the potential well. Or it may transfer energy back to the thermal bath via viscosity and go down in the potential. These energy transfers with the thermal bath constitute heat $\hat{Q}$ and, and since this energy can be stored as potential energy, this is the internal energy $\hat{E}$ of the particle. In our system then the internal energy is $\hat{E}=V_{\kappa ,x_{0}}=\frac{1}{2}\kappa {\left(x-x_{0}\right)}^{2}$ [17,18,19,20]. We will use $\hat{a}$ to denote a stochastic variable and the regular letter *a* for the average over realizations, i.e., $E=\langle \hat{E}\rangle$. Another form of energy transfer is work $\hat{W}$: an external agent may modify the harmonic potential (changing the parameters) and increase or decrease the potential energy of the particle. If the internal energy depends on a parameter λ that is modified from $\lambda_{0}$ to $\lambda_{f}$ then, formally, the definition of work is:(2)$$\hat{W}=\int_{\lambda_{0}}^{\lambda_{f}}\frac{\partial E}{\partial \lambda }d\;\lambda$$

This is best seen with an example. For instance, consider a fast increase of the stiffness of the potential from $k_{i}$ to $k_{f}$. If the increase is very fast, so that the particle does not modify its position *x* during the time, the stiffness is changing, the energy of the particle increases by an amount $\Delta V=\frac{1}{2}{\left(x-x_{0}\right)}^{2}\left(k_{f}-k_{i}\right)$. This energy is supplied by the agent controlling the potential who has then performed a work $\hat{W}=\Delta V>0$. Consequently, the particle is in a tighter parabola and the equilibrium dispersion of the position of the particle decreases, so that this is commonly referred to as a compression. If the stiffness is decreased, work is exerted on the agent by the system and, since the distribution of particle positions will eventually widen, this corresponds to an expansion.

With these definitions, energy is conserved and the first law is fulfilled either at the level of trajectories $\hat{\Delta }E=\hat{Q}+\hat{W}$ or as averages E=Q+W [14,15,16,17,18,19,20].

In order to extract energy from the thermal bath, we propose the following cyclic operation in two stages:1.Confinement of the Brownian particle by (repeated) measuring and feedback2.Isothermal expansion

The system works as a motor if the work obtained in the isothermal expansion exceeds the work that is needed for confinement. During a compression, the free energy of the system increases (due to the entropy decrease). Using reversible feedback confinement [2], we can minimize the work that is needed for stage 1, which turns out to vanish, and extract all of the free energy increase of stage 1 as work during stage 2. Let us analyze each stage in more detail.

### 2.1. Optimal Confinement

The confinement of a system to a small region of the phase space (at constant temperature) implies a decrease of entropy of the system. For the entropy of a Brownian particle, we use the standard choice of Shannon’s entropy, S=−k∫ρ(x)log(x)dx, where ρ(x) is the probability distribution of the particle position. With this choice, the second law of thermodynamics is fulfilled on average and the thermodynamic relation F=E−TS is recovered for a system in contact with a thermal bath. Although, strictly speaking, this is a generalization of the free energy to non-equilibrium systems, in systems that are in contact to a thermal bath it plays a similar role as the standard thermodynamic free energy, and stochastic thermodynamics for our system closely resembles macroscopic thermodynamics [9].

Let us consider, for simplicity, that the internal energy change between the initial and final states of the confinement process vanishes (we will see later that this is the case in our particular system). A reduction of entropy then corresponds to an increase of free energy ΔF=ΔE−TΔS. This increase in free energy could then be extracted as work in an isothermal expansion. However, the whole process cannot operate as a motor, as this will defeat the second law (extracting work from a single thermal bath). Indeed, the second law states for the confinement
(3)$$W_{1}\geq \Delta F_{1}\;(w/o.\;\mathrm{feedback})$$
and then for the isothermal expansion back to the initial state ($\Delta F_{2}=-\Delta F_{1}$)
(4)$$W_{2}\geq \Delta F_{2}=-\Delta F_{1}\;(w/o.\;\mathrm{feedback})$$
so that $W_{\mathrm{total}}=W_{1}+W_{2}\geq 0$ and the system dissipates energy into the thermal bath.

However, as explained above, when measuring and feeding back to the system, *W* is bounded by (Equation 1) instead. Thus, the work that is needed for the confinement may be reduced and the work output of the cyclic process ($\Delta F_{\mathrm{cycle}}=0$) may be negative:(5)$$W_{\mathrm{total}}\geq -kTI$$
Notice that mutual information is always a positive quantity.

Following [2], we propose a reversible feedback confinement that can confine the particle with $W_{1}=0$ and, as will be shown later (see Equation (Equation 13)), without dissipating heat to the thermal bath, so that the increase in free energy that is produced by the confinement can later be completely recovered as work during a quasistatic expansion in stage 2.

For a system that is in contact with a thermal bath, a feedback process is reversible if the Hamiltonian is modified after the measurement, so that probability of the state of the system conditioned on the measurement outcome is the Gibbsian state of the new Hamiltonian. After a measurement, the probability to find a given state changes instantaneously, the new probability distribution takes into account the information obtained, and ut must be updated according to Bayesian inference. If the Hamiltonian also changes rapidly and the Gibbs state of the new Hamiltonian matches the posterior probability distribution, the system remains at equilibrium and no further evolution of the probability distribution ensues until a new measurement is taken.

In our model, we take the common assumption of Gaussian measurement errors. If the particle is located at a position *x*, then the measurement outcome *m* is Gaussian distributed around *x* and the dispersion $\sigma_{m}$ quantifies the measurement error:(6)$$q\left(m|x\right)=\frac{1}{\sqrt{2\pi \sigma_{m}^{2}}}e^{-{\left(m-x\right)}^{2}/2\sigma_{m}^{2}}$$
After a measurement, the probability distribution of the position of the particle updates according to Bayes’ theorem from the initial distribution ρ:(7)$$\rho^{\prime }\left(x|m\right)=\frac{\rho (x)q(m|x)}{\pi (m)}$$
where π(m)=∫dxq(m|x)ρ(x) is the marginal distribution of the measurement outcome.

For a Brownian particle in a time-independent potential, the equilibrium distribution is its corresponding Gibbs distribution:(8)$$\rho \left(x\right)\propto e^{-V(x)/kT}$$
In a harmonic potential, it is a Gaussian, centered in the trap position $x_{0}$ and with variance being given by $\sigma^{2}=kT/\kappa$. It can be shown [7] that, after a measurement, the new distribution that is computed according to (Equation 7) remains Gaussian. If the initial distribution has mean $\bar{x}$ and standard deviation σ, after a measurement, then the distribution updates to a Gaussian with the mean and deviation given by [2]:
(9)$$\begin{array}{cc} {\bar{x}}^{\prime }\left(m\right) & =\frac{\sigma_{m}^{2}}{\sigma^{2}+\sigma_{m}^{2}}\bar{x}+\frac{\sigma^{2}}{\sigma^{2}+\sigma_{m}^{2}}m \end{array}$$(10)$$\begin{array}{cc} \frac{1}{{\sigma \prime }^{2}} & =\frac{1}{\sigma^{2}}+\frac{1}{\sigma_{m}^{2}} \end{array}$$

We can make the post-measurement distribution an equilibrium distribution by setting a new center of the trap position $x_{0}^{\prime }$ and stiffness $\kappa^{\prime }$, as
(11)$$\begin{array}{cc} \kappa^{\prime } & =kT/{\sigma \prime }^{2} \end{array}$$
(12)$$\begin{array}{cc} x_{0}^{\prime }\left(m\right) & ={\bar{x}}^{\prime }\left(m\right) \end{array}$$

Notice that $\kappa^{\prime }>\kappa$; hence, the particle is more tightly bound or confined after this change. Additionally, Equation (10) implies $\sigma^{\prime }<\sigma$, so that every measurement and feedback step further reduces the variance of the particle distribution.

In order to check this reduction of variance in simulation, we have computed the particle distribution after a measurement. For this, we first generate a large number of trajectories, starting from an initial equilibrium distribution for a harmonic potential centered in position $x_{0}=0$ and corresponding dispersion σ=3.0, as depicted in Figure 1(left). After some time interval, for each trajectory, we measure its position by generating a (Gaussian) random measurement outcome *m* around the actual position *x* with dispersion $\sigma_{m}=3.0$ (see the details in Section 5). We can then fix a small interval around a given measurement of the position (m,m+Δm) of our choice, for instance (0.89, 0.98), and only check the realizations that gave a measurement in that interval. The distribution of the actual positions *x* of these particular realizations are distributed as in (Equation 7). In our case, a Gaussian with new reduced standard deviation $\sigma^{\prime }=2.12$ is given by (10). This can be seen in Figure 1 (right).

This process of measurement and feedback can be repeated and a new, more confined state could be achieved. Figure 2 (top) shows the confining effect of repeating this procedure.

In every measurement and feedback step, the trapped Brownian particle stays in equilibrium with the thermal bath at temperature *T*. Consequently, the average energy is not modified by the feedback process. The average internal energy *E* of a trapped particle in one dimension is given by the equipartition theorem, as E=kT/2. Because the process is isothermal, ΔE=0. On the other hand, always being in equilibrium, there is no relaxation of the particle distribution and the heat that is transferred from the heat bath vanishes on average Q=0. Therefore, according to the first law, the average work done on the system also vanishes:(13)$$\Delta E_{1}=Q_{1}+W_{1}=0\Rightarrow W_{1}=-Q_{1}=0$$

This has been checked in simulations, as shown in Figure 2 (bottom). Details about work computation during measurement and feedback can be found in Section 5.

In general, for other feedback protocols where the stiffness of the trap is suddenly changed, work is performed, on average [21], as in the simple example described after Equation (Equation 2). The feedback process used here is different (in addition to a sudden increase of stiffness, trap position is also modified in a precisely combined manner) and it is special in the sense that average work vanishes. As encoded in Equation (Equation 1), this can solely be achieved by using information regarding the position through measurement in the feedback (see Equation (Equation 9) for the new trap position). To see why this matters, consider our Brownian particle in a harmonic potential, where the observer happens to know that the particle is exactly at the bottom of the well. This would allow for this external agent to increase the stiffness of the potential well with an abrupt change, without performing work, since the energy of the particle is always zero at the bottom of the well, for any stiffness. The confining protocol is a refinement of this idea that works for any position of the particle, by displacing the bottom of the potential well towards the measured particle position and changing the stiffness in a suitable manner.

Furthermore, one can compute the mutual information that is obtained in the process of measurement and evaluate the increase in free energy $\Delta F_{1}$ for the confinement stage. From the definition of mutual information:(14)$$I\left(m,x\right)=\int \pi \left(m\right)q\left(m|x\right)\log\frac{q(m|x)}{\pi (m)}$$

When considering that the measurement outcome distribution q(m|x) and the marginal distribution π(m) are Gaussian with variance $\sigma_{m}^{2}$ and $\sigma^{2}+\sigma_{m}^{2}=\sigma_{m}^{2}\sigma^{2}/{\sigma \prime }^{2}$, respectively, the information that is acquired in a measurement is
(15)$$I\left(m,x\right)=-\frac{1}{2}\log\frac{{\sigma \prime }^{2}}{\sigma^{2}}\geq 0$$
Mutual information intuitively measures the decrease in uncertainty of variable *x* if we know the value of *m*, or vice versa [22]. In our case, from (10), if the measurement error $\sigma_{m}$ is very large then $\sigma^{\prime }\approx \sigma$ and we extract almost no information from measuring (I≈0). Conversely, for infinite precise measurement $\sigma_{m}\to 0$, then $\sigma^{\prime }\to 0$, and we obtain infinite information from a measurement, as an infinite precise description of a position would require an infinite number of bits to store it.

The entropy of a Gaussian of variance $\sigma^{2}$ is $S\left(\rho \right)=k\log\sigma \sqrt{2\pi e}$. In the measurement process, the distribution changes from a Gaussian of variance $\sigma^{2}$ to a Gaussian of variance ${\sigma \prime }^{2}$, and we have
(16)$$\Delta S_{1\mathrm{step}}=k\log\sigma^{\prime }\sqrt{2\pi e}-k\log\sigma \sqrt{2\pi e}=k\frac{1}{2}\log\frac{{\sigma \prime }^{2}}{\sigma^{2}}=-kI\left(m,x\right)$$
Because ΔE=0, we finally obtain
(17)$$\Delta F_{1\mathrm{step}}=\Delta E-T\Delta S_{1\mathrm{step}}=kTI\left(m,x\right).$$
This is valid for every measurement and feedback step while using the reversible feedback protocol. In a sequence of confinement steps with successive variances $\sigma_{0},\sigma_{1},\ldots ,\sigma_{n}$, the total information is
(18)$$I_{\mathrm{total}}=-\frac{1}{2}\sum_{i=1}^{n}\log\frac{\sigma_{i}^{2}}{\sigma_{i-1}^{2}}=-\frac{1}{2}\log\frac{\sigma_{n}^{2}}{\sigma_{0}^{2}}.$$
$\sigma_{n}^{2}$ can be obtained from (10) by recursion, giving:(19)$$\frac{1}{\sigma_{n}^{2}}=\frac{1}{\sigma_{0}^{2}}+n\left(\frac{1}{\sigma_{m}^{2}}\right)$$

Finally, the free energy difference between the final and initial states in the confinement stage is
(20)$$\Delta F_{1}=kTI_{\mathrm{total}}=\frac{kT}{2}\log\frac{\sigma_{0}^{2}}{\sigma_{n}^{2}}=kTI_{\mathrm{total}}.$$

Every bit of information that is extracted in the measurement is turned into an increase of free energy during the confinement stage and it can be converted into useful work in the subsequent expansion.

### 2.2. Work Extraction by Isothermal Expansion

If an external agent changes the stiffness of the optical trap from $\kappa_{i}$ to $\kappa_{f}<\kappa_{i}$, energy is recovered as work, as explained above. In a quasistatic process, the work done by the system is given by the free energy difference. Because stage 2 completes the cycle of operation of the motor ending in the initial state, we have $\Delta F_{2}=-\Delta F_{1}$ and
(21)$$W_{\mathrm{total}}=W_{1}+W_{2}=0-\Delta F_{1}=-kTI_{\mathrm{total}},$$
which corresponds to extracted work. In fact, it saturates expression (Equation 5) and it is the maximum possible work that can be extracted while using the information that was obtained in the measurements.

This result can also be recovered by the direct computation of the work of a process changing stiffness from $\kappa_{i}$ to $\kappa_{f}$ and while taking into account that, for a quasistatic process, one can use the equipartition theorem stating $\langle x^{2}\rangle =kT/\kappa \left(t\right)$, with κ(t) the instantaneous value of the stiffness. Subsequently, the average work during the expansion, according to (Equation 2), reads:(22)$$W_{2}=\int_{\kappa_{i}}^{\kappa_{f}}d\kappa \frac{\langle x^{2}\rangle }{2}=\frac{kT}{2}\int_{\kappa_{i}}^{\kappa_{f}}\frac{d\;\kappa }{\kappa }=\frac{kT}{2}\log\frac{\kappa_{f}}{\kappa_{i}}$$
The expansion starts at the end of the confinement process with a distribution of variance $\sigma_{n}$ and ends at $\sigma_{0}$. Subsequently, while using the relation between stiffness and variance in the confinement stage (Equation 11), we have
(23)$$W_{2}=\frac{kT}{2}\log\frac{\kappa_{f}}{\kappa_{i}}=-\frac{kT}{2}\log\frac{\sigma_{n}^{2}}{\sigma_{0}^{2}}=-kTI_{\mathrm{total}}$$
Notice that during both the confinement and expansion the system must be at equilibrium in order to transform every bit of information into useful work.

In practice, though, for a process changing the stiffness of the potential to be approximately quasistatic, it is enough that the time of the process is large compared to the inverse frequency of the trap given by ν=κ/γ. This is the criterion that we have used for simulations. Additionally, it is worth noting that, even though the work in every realization of the expansion may differ in principle in a stochastic system, work is—in this particular example—a self-averaging quantity: for a quasistatic expansion, the total work obtained in any realization is very similar to its average value. The argument for self-averaging of the work is the following: from work definition (Equation 2), work in a single realization when expanding is $\hat{W}=\int \frac{x^{2}}{2}d\kappa$. If the expansion is very slow, in the time κ is modified a certain small amount, the particle position has time to fluctuate and sample the whole quasi-equilibrium distribution and $x^{2}$ approximately can be replaced by its average value (see the full computation in [14]).

Figure 3 depicts the complete diagram of the proposed cycle.

Finally, one could also define an efficiency η as the ratio between the extracted thermodynamic resource (work) and the thermodynamic resource consumed to make the engine run, in this case information. With this definition, this reversible feedback engine attains the maximum efficiency:(24)$$\eta =\frac{-W}{kTI}=1$$
as in a similar system [23], with just one measurement per cycle.

## 3. Results

### 3.1. Work Is Optimal

We have performed computer simulations of the model system that is described above. Figure 4 depicts part of two consecutive cycles, each of them with a confinement stage that is composed of 10 measurement and feedback steps, followed by an isothermal expansion. The top panel depicts the particle position (gray line), trap center (blue line), and measurement outcomes (red dots), whereas the bottom panel shows the evolution of the stiffness along the cycle.

Figure 5 shows the cumulative work that was done on the system along the time of a single cycle. The thick solid line represents the average over 200 cycles. Every cycle consists of a confinement that is achieved by measuring the particle position 10 times and the subsequent isothermal expansion. Average work extracted (W<0) by the end of the cycle approaches the expected result that is given by Equations (Equation 18), (Equation 19) and (Equation 21), marked with dashed black line. The shaded area represents the variance of the work, which is substantially large. As is apparent from the figure, most of the variance comes from the confinement step, with the quasistatic work being a self-averaging quantity. Finally, work that corresponds to two particular cycles is shown by thin blue lines.

### 3.2. Power and Efficiency with Higher Measurement Errors

Consider two setups, *A* and *B*, with different measurement errors being given by variances $\sigma_{mA}^{2}$ and $\sigma_{mB}^{2}=2\sigma_{mA}^{2}$. Suppose that only one measurement step is performed in each system before the expansion. According to our discussion above, the measurement information that can be later transformed into work is smaller in system *B* than in *A*:(25)$$I_{B1}=\frac{1}{2}\log\frac{\sigma_{mB}^{2}+\sigma_{0}^{2}}{\sigma_{mB}^{2}}=\frac{1}{2}\log\left(1+\frac{\sigma_{0}^{2}}{2\sigma_{mA}^{2}}\right)<\frac{1}{2}\log\left(1+\frac{\sigma_{0}^{2}}{\sigma_{mA}^{2}}\right)=I_{A1}$$

However, we can obtain as much information in system *B* with two measurements as in system *A* with one measurement. After two measurements, while using the reversible confinement protocol, the variance of the equilibrium distribution $\sigma_{B2}^{2}$ in system *B* is equal to the variance in system *A* after one measurement $\sigma_{A1}^{2}$:(26)$$\frac{1}{\sigma_{B2}^{2}}=\frac{1}{\sigma_{0}^{2}}+2\left(\frac{1}{\sigma_{mB}^{2}}\right)=\frac{1}{\sigma_{0}^{2}}+\left(\frac{1}{\sigma_{mA}^{2}}\right)=\frac{1}{\sigma_{A1}^{2}}$$
Using (Equation 18), we obtain:(27)$$I_{B}=\frac{1}{2}\log\frac{\sigma_{B2}^{2}}{\sigma_{0}^{2}}=\frac{1}{2}\log\frac{\sigma_{A1}^{2}}{\sigma_{0}^{2}}=I_{A}$$

As explained above, this implies that the same work can be extracted in the subsequent quasistatic expansion. In fact, bothof the systems run with the same efficiency η=1; hence, every bit of information is turned into work in the expansion. Furthermore, system *B* can also be run in principle at the same power as system *A*. During the confinement process, after the adjustment of the potential, the particle distribution is at equilibrium. No relaxation occurs, as explained previously. Therefore, a new measurement and feedback step could, in principle, be taken immediately after, in rapid succession. Thus, halving the time between measurements in system *B* as compared to system *A* ensures the same cycle time. As the work obtained is also the same, both of the systems operate with the same power. Figure 6 depicts this, where we show the simulation results for system *A* with one measurement and expansion and system *B* with two (faster) measurements and expansion. Approximately the same work is obtained in both systems. For reference, we have also marked the expected extracted work for a system with tge measurement error given by $\sigma_{mB}$, but using just one measurement.

## 4. Discussion

Reversible feedback confinement is an optimal way of reducing the entropy of a system to be later used for work extraction. Nevertheless, it requires a high degree of control over the Hamiltonian, to adapt it to the new probabilistic state after the measurement. This might be a limitation for experimental realizations, although a low dissipation is expected, even if a similar or approximate protocol is implemented. Theoretically, the dissipation could be accounted for by using the Kullback–Leibler distance between the post-measurement particle distribution and the equilibrium distribution of the potential after feedback [24], if they were different due to a less precise tuning of the potential.

In principle, for a measurement and feedback protocol, imprecision in the measurement, which will inevitably arise in an experimental setup, will limit the work extraction or power. Nevertheless, we have shown here that this limitation can be overcome by adding more measurement steps before the quasistatic expansion, as long as the reversible feedback confinement protocol is used. In principle, the application of this protocol is instantaneous. In practice, this means that the confinement may be applied in a very short time, limited maybe by the response time of the feedback mechanism or the measurement acquisition time. Thus, if the response times of measurement, feedback, and Hamiltonian modification are fast as compared to system’s relaxation time, optimal work extraction is feasible, even with a high degree of inaccuracy in the measurement, while using repeated optimal feedback.

## 5. Materials and Methods

The confined Brownian particle evolves according to Langevin equation:(28)$$\gamma \dot{x}=-V_{\kappa ,x0}^{\prime }\left(x\right)+\xi (t),$$
with ξ(t) Gaussian white noise $\langle \xi \left(t\right)\xi \left(t^{\prime }\right)\rangle =2kT\gamma \delta \left(t-t^{\prime }\right)$, *T* bath temperature and *k* Boltzmann’s constant. The potential $V_{\kappa ,x0}\left(x\right)$ is defined above and it is controlled through measurement and feedback. Model simulations were performed in C language, solving the Langevin evolution equation with the Heun method for a stochastic differential equation [25]. We provide, in the following, some details on work computation, measurement, and feedback steps. For full details, the code is available here: http://seneca.fis.ucm.es/ldinis/code/extract_optimal_work.zip.

Measurement. In order to perform a measurement in the simulation, a Gaussian number “r” with zero average and standard deviation 1 is generated. Subsequently, if particle position is *x*, the measurement outcome *m* is
(29)$$m=x+\sigma_{m}r$$Notice that *m* is then distributed according to Equation (Equation 6)Feedback. Immediately after measurement, and using the measurement outcome *m* just computed, the potential parameters κ and $x_{0}$ are recomputed according to Equations (Equation 9) to (12). Notice that the old values need to be stored in an auxiliary variable for the work computation, as explained in the following step.Work computation during feedback process. According to its definition for a trajectory, work is the difference in the potential energy when the potential is changed. If $\kappa \to \kappa^{\prime }$ and $x_{0}\to x_{0}^{\prime }$ as a result of measurement and feedback, then work is computed as
(30)$$\Delta W=\frac{1}{2}\kappa^{\prime }{\left(x-x_{0}^{\prime }\right)}^{2}-\frac{1}{2}\kappa {\left(x-x_{0}\right)}^{2}$$This ΔW is added to a variable *W* that stores the cumulative work that was done along the whole simulation.After the feedback, evolution equation resumes with the new potential parameters.Work during expansion. Work is also performed as a result of the change in κ during an expansion. In the simulation, during the expansion stage, κ changes an amount $\Delta \kappa =\frac{\kappa_{f}-\kappa_{i}}{Nexp}$ in every time step, where Nexp is the number of time steps of the expansion. Therefore, in a time step, a work
(31)$$\Delta W=\frac{1}{2}\Delta \kappa {\left(x-x_{0}\right)}^{2}$$
is performed. Again, this ΔW has to be added to the variable *W*, which stores the total or cumulative work of the whole process.

## Figures and Tables

**Figure 1 entropy-23-00008-f001:**
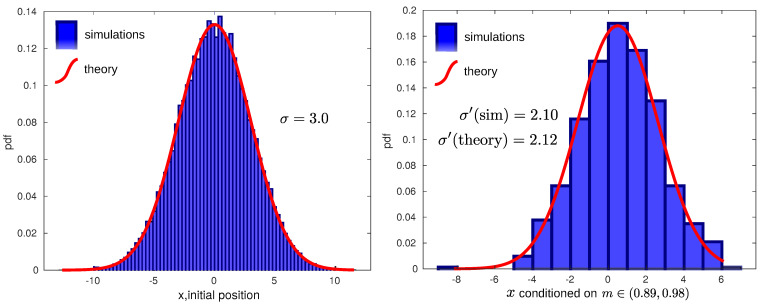
Reduction of variance after measuring. **Left**: Initial distribution. Histogram of 40,000 random Gaussian numbers centered in 0 with standard deviation σ=3.0 (blue bars) and a theoretical Gaussian distribution with the same parameters (red continuous line). **Right**: Posterior distribution. Histogram of particle positions with measurement outcomes in a given interval (0.89, 0.98) (blue bars) and prediction according to Bayes’ theorem (Equation 7) (red). Measurement outcomes were performed with measurement error $\sigma_{m}=3.0$. Using $\sigma_{m}=3.0$ and σ=3.0 in (10) gives $\sigma^{\prime }=2.12$, which matches the sample standard deviation of 2.10.

**Figure 2 entropy-23-00008-f002:**
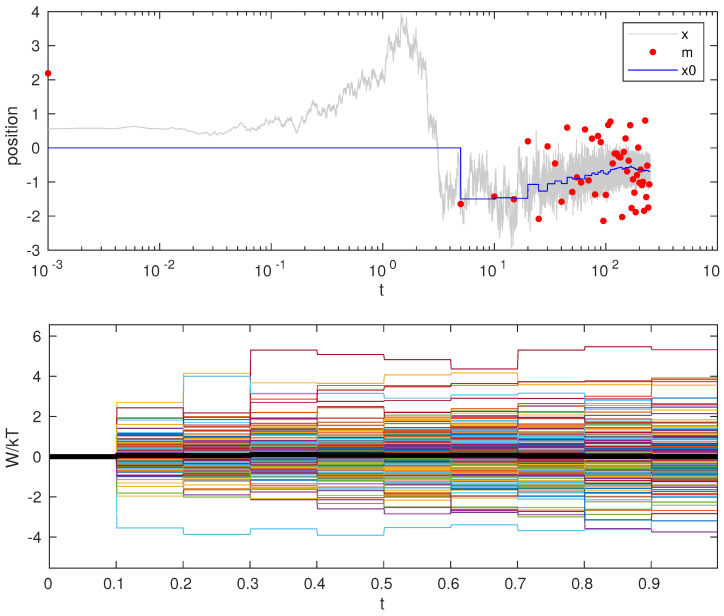
Confinement. **Top**: particle trajectory (gray line), measurement outcome (red dots) and trap center position (blue line). **Bottom**: Cumulative work for different realizations (color lines) and its average over 200 realizations (thick black line) for confinement in 10 measurement steps. See the simulation details in Section 5. Initial trap stiffness κ=0.1 and position $x_{0}=0$. Initial condition is equilibrium with trap potential to avoid transient due to equilibration. Particle diffusion coefficient D=1 and friction γ=1.

**Figure 3 entropy-23-00008-f003:**
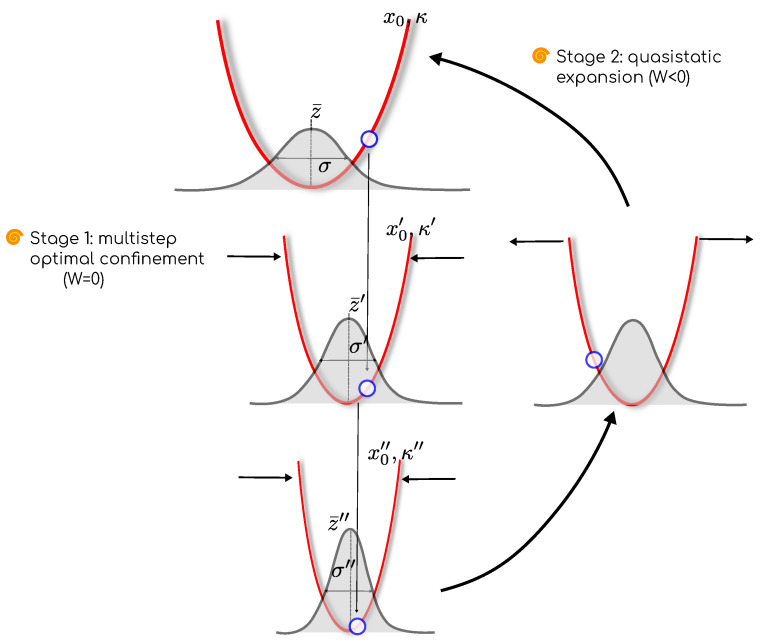
Cycle for extracting work from a thermal bath with inaccurate measurements.

**Figure 4 entropy-23-00008-f004:**
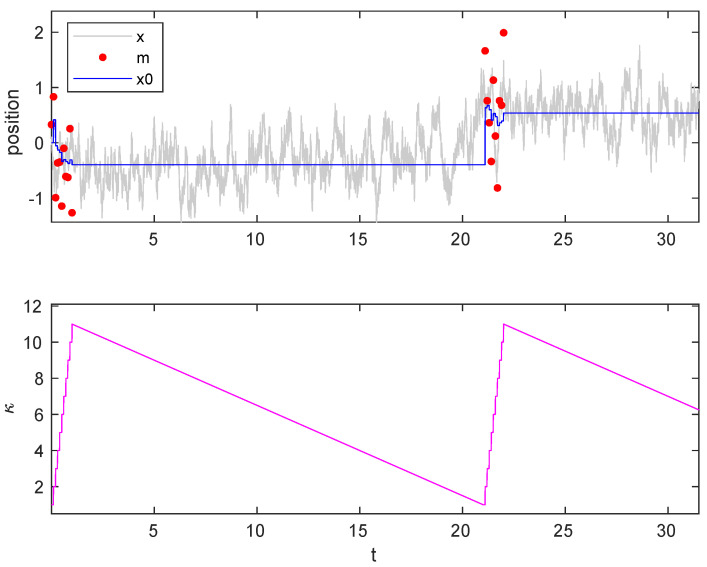
Trajectories. (**Top**) Particle trajectory (gray continuous line), trap center (blue continuous line), measurement outcomes (red dots). (**Bottom**) Stiffness evolution during the cycle. Every cycle starts with κ=1, there are 10 measurement steps, followed by quasistatic expansion. D=1, γ=1.

**Figure 5 entropy-23-00008-f005:**
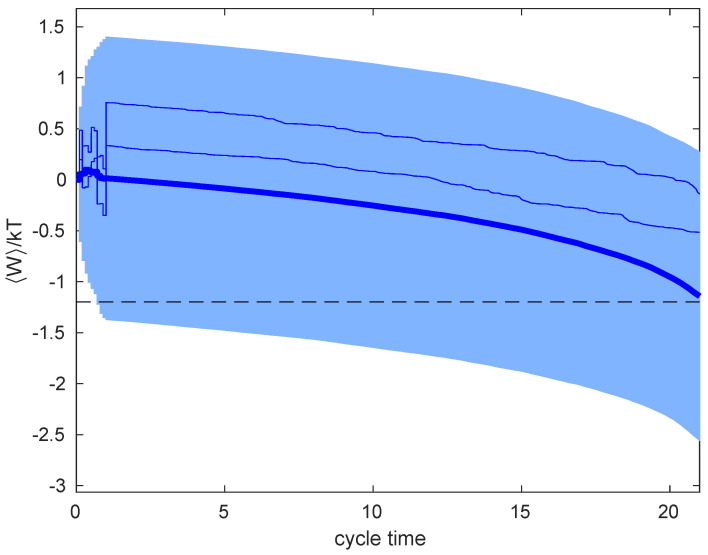
Average cumulative work along the confinement-expansion cycle (thick blue line) computed from 200 realizations. The shaded area corresponds to one standard deviation from the average. Thin blue lines represent cumulative in two representative cycles. Simulation parameters are: Δt=0.001, time between measurements τ=0.1, number of measurements before expansion is 10, measurement error $\sigma_{m}^{2}=1$, initial stiffness of the trap κ=1, diffusion coefficient D=kT/γ=1, and drag coefficient γ=1.

**Figure 6 entropy-23-00008-f006:**
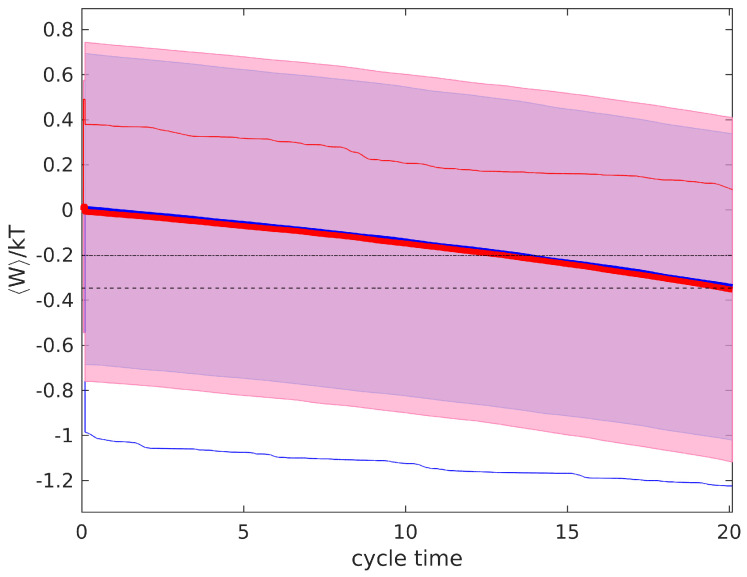
Average work extraction for two different measurement errors, using one measurement with variance $\sigma_{mA}^{2}=1$ (blue thick line), and using two measurements with variance $\sigma_{mB}^{2}=2$ (red thick line). Dashed line represents expected work extraction and fine dashed line corresponds to expected work extraction with just 1 measurement of variance $\sigma_{mB}$. Thin lines represent single realizations of the work in system *A* (blue) and *B* (red).

## Data Availability

Data for this study was generated using custom computer code. The code and instructions are available at http://seneca.fis.ucm.es/ldinis/code/extract_optimal_work.zip.

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
