# Peer review of "Extracting Work Optimally with Imprecise Measurements"

_entropy, 2020, doi:10.3390/e23010008_

Round 1
Reviewer 1 Report
The present manuscript discusses the application of reversible feedback confinement to the optimal work extraction of an overdamped Brownian particle. The results are interesting. Nevertheless, a few points need some clarification before accepting the manuscript for publication:
- Between Eqs.(9) and (10), the authors say that the post-measurement distribution can be made an equilibrium distribution by setting a new center of the trap position and stiffness. Is there a simple way of understanding why the process of setting new values of x_0 and \kappa has zero cost on average? For instance, in Phys. Rev. Lett. 98, 108301 (2007), it is shown (see Eq.(20)) that there is a non-zero average work performed when the stiffness is abruptly changed.
- In line 21, the authors say that the quantity I appearing in Eq.(1) "is the information gained in the measurement". However, for infinite precision, \sigma_m would go to zero and q(m|x) in Eq.(5) would approach a Dirac-delta function. In this case, the mutual information I given by Eq.(14) seems to go to zero. Is this correct? Would an infinitely precise measurement lead to no gained information?
- In line 99, the authors say that "in a stochastic system, work is a self-averaging quantity and for a quasistatic expansion the total work obtained in realizations is very similar to the average". Although the authors refer to Ref.[14] about it, what does the concept of self-averaging really mean in this case? What do the authors mean about it? In Phys. Rev. E 75, 021116 (2007), the work distributions of an ideal gas driven by an adiabatic and quasistatic expansion/compression are obtained. These distribuitions show a finite width, which means that single realizations are not very similar to the average value in general. Why is this not happening in the model of the present manuscript?
- In Section 3, the authors should give more details about how the feedback process is implemented in the simulations. For instance, what is one really doing when the position is measured? For instance, if there is an ensemble of realizations, does measuring the position simply mean recording the position of each member of the ensemble at the measurement time? If this is the case, how would this change the spreading of the ensemble? Could the authors discuss this in detail and show simulations confirming the change in the variance after the position measurement?
- About the simulations, in Section 5, it is said that model simulations were performed solving Langevin evolution. However, it seems that the numerical solution of Langevin dynamics had to be combined with other numerical methods implementing the feedback process. Is that correct?
- Typos: in line 71, the relation between \sigma and \kappa; in line 110, "s ingle".
Reviewer 2 Report
See attached file.

Round 2
Reviewer 1 Report
The authors have answered all the questions which were previously posted satisfactorily. I recommend the manuscript for publication as it is.
Author Response
We thank the referee for previous comments that have improved the paper. No action needed this time so we just checked the manuscript for typos and errors. All changes marked in red.
Reviewer 2 Report
See attached file.
